# Synthesis of CuAl-LDHs by Co-Precipitation and Mechanochemical Methods and Selective Hydrogenation Catalysts Based on Them

Olga B. Belskaya [1,*] , Elena N. Terekhova [1], Oksana V. Gorbunova [1], Ivan V. Muromtsev [1], Mikhail V. Trenikhin [1], Aleksei N. Salanov [2] and Vladimir A. Likholobov [2]

[1] Center of New Chemical Technologies BIC, Boreskov Institute of Catalysis, 54 Neftezavodskaya Street, 644040 Omsk, Russia; e.terechova@ihcp.ru (E.N.T.); oxana_gorbunova@inbox.ru (O.V.G.); muromtseviv@gmail.com (I.V.M.); tremv@yandex.ru (M.V.T.)

[2] Boreskov Institute of Catalysis, 5 Acad. Lavrentieva Ave., 630090 Novosibirsk, Russia; salanov@catalysis.ru (A.N.S.); likholobov47@mail.ru (V.A.L.)

[*] Correspondence: obelska@ihcp.ru; Tel.: +7-(3812)-670-474

**Abstract:** The paper presents the results of the synthesis and study of CuAl layered double hydroxides (LDHs) as well as their application as catalysts for the selective hydrogenation of crotonaldehyde. Phase-homogeneous LDHs were obtained by co-precipitation and mechanochemical methods, and critical parameters ensuring the formation of the target product were identified. In the case of coprecipitation, the formation of LDH is most affected by the pH of the reaction medium and the $CO_3^{2-}/Al^{3+}$ ratio. The optimal $CO_3^{2-}/Al^{3+}$ ratio is ca. 0.5–0.8 and pH 9.5–10.0. When mechanochemical synthesis is used, at 500 m·s$^{-2}$ and 60 min, it is possible to obtain a single-phase CuAl LDH, whereas at higher energies, LDH is destroyed. The mechanochemical method makes it possible not only to reduce the synthesis time and the amount of alkaline wash water but also to obtain more dispersed copper particles with a higher hydrogenating activity. The conversion of 2-butenal (T = 80 °C, P = 0.5 MPa, 180 min, ethanol) for this sample was 99.9%, in contrast to 50.5% for the catalyst obtained by co-precipitation. It is important that, regardless of the conversion, both catalysts showed high selectivity (S = 90–95%) for the double bond hydrogenation.

**Keywords:** layered double hydroxides; copper catalyst; co-precipitation; mechanochemical methods; crotonaldehyde hydrogenation

## 1. Introduction

One of the current trends in the development of catalysts is the production of active catalytic compositions based on non-noble metals. Therewith, an efficient approach to ensuring a uniform distribution and high dispersion of the active metal is the use of layered double hydroxides (LDHs) as catalyst precursors, in which the cation of this metal is embedded in the structure of bi- or polymetallic hydroxide layers.

Layered double hydroxides (LDHs) are inorganic compounds with a hydrotalcite-like structure consisting of positively charged octahedral layers of di- and trivalent metals (Me(II) and Me(III)) and negatively charged anions in the interlayer space ($A^-$), which compensate for the positive charge. LDHs have the general formula $[M(II)_{1-x}M(III)_x(OH)_2][A^{n-}]_{x/n}·mH_2O$, where *M* is metal cations, *A* is the interlayer anion, *x* is the molar ratio $M^{3+}/(M^{2+} + M^{3+})$, and *m* is the number of water molecules. LDHs are analogs of the natural hydrotalcite mineral $Mg_6Al_2(OH)_{16}CO_3·4H_2O$, but the development of synthetic methods made it possible to replace $Mg^{2+}$ with bivalent cations ($Co^{2+}$, $Ni^{2+}$, $Cu^{2+}$, $Zn^{2+}$, $Cd^{2+}$, $Ca^{2+}$), and $Al^{3+}$ with tri– and even tetravalent cations having close ionic radii, for example, $In^{3+}$, $Ga^{3+}$, $Au^{3+}$, $Cr^{3+}$, $V^{3+}$, $Ce^{3+}$, $Sn^{4+}$, $Zr^{4+}$ [1–4]. As a result, the application field of LDH-based materials was expanded to use them as catalysts or catalyst supports for various processes.

Thus, catalysts based on layered double hydroxides in which the magnesium cation is partially or completely replaced by the nickel or iron cation are efficient for producing hydrogen by decomposition of ammonia [5,6]. Bi- and polymetallic (Ni, Co, Zn, Mg)Al- LDHs can also be used in the synthesis of catalysts for dry reforming reactions [7], aqueous-phase hydrogenation of furfural [1,3], photochemical [8], and electrochemical [9,10] reactions. Thus, multicomponent systems containing CoFe-LDH [9] and NiSe@NiFe-LDH [10] were used as electrocatalysts for the hydrogen evolution reaction.

Of particular interest are copper-containing catalysts of this type due to the mild conditions of metal reduction, low price, availability, and good selectivity when used for the reduction of nitroarenes, reductive amination of aldehydes or ketones, selective hydrogenation of the formyl group in furan aldehydes, and n-methylation of p-anisidine [11–13]. In addition, copper-containing catalysts based on polymetallic compositions CuFe-, CuZnGa-, CuMgAl-, and CuNiSn-LDH demonstrate good catalytic performance in the oxidation of 5-hydroxymethylfurfural to 5-formyl-2-furanic acid [14], production of methanol from syngas [15], single-step liquid-phase hydroxylation of phenol to catechol and hydroquinone [16], and decomposition of phenol [17].

Similar to LDHs of other compositions, CuAl-LDH is produced mostly by the co-precipitation method, which includes two main steps: (i) precipitation of hexaaqua metal complexes in an alkaline medium leading to the formation of brucite-like layers with uniformly distributed cations and solvated interlayer anions in them; and (ii) aging of the mixture to improve the crystallinity of the precipitate. Despite the simplicity of the hardware design, the obvious disadvantages of this method are its duration and formation of a difficult-to-filter gel-like product, as well as the necessity of utilizing a large amount of alkaline wash water, which makes the method environmentally unsafe. It should be noted that co-precipitation commonly ensures high purity of the obtained LDHs of various compositions and reproducibility of their structure and properties [17,18]. At the same time, it was shown that, unlike MgAl-LDH, it is not easy to obtain CuAl-LDH, even though the ionic radii of $Mg^{2+}$ and $Cu^{2+}$ are close and equal to 0.72 A° and 0.73 A°, respectively. The published results concerning the synthesis of copper-containing LDHs are contradictory [19–23]. In many studies, the co-precipitation method did not allow obtaining a single-phase product despite the variation in the Cu/Al ratio and synthesis conditions; $Na_2Al_2(CO_3)_2$, $Cu_2(NO_3)(OH)_3$, $Cu_2(CO_3)(OH)_2$, $Na_2Cu(CO_3)_2$, and $Al(OH)_3$ were present as impurities. It was demonstrated [24] that difficulties with the synthesis of CuAl-LDH have two reasons: its formation is determined by various factors, such as the nature of the initial compounds, the pH of precipitation, the Cu/Al molar ratio, the concentration and composition of the precipitator, and aging conditions; the result of the synthesis differs significantly depending on the combination of selected parameters.

The expansion of the CuAl-LDHs application field and the need to vary their structure, along with the optimization of the conditions of the synthesis by co-precipitation, initiate the development of more efficient and reproducible synthesis methods [25]. Among them is the mechanochemical method, which implies a solid-phase reaction between the initial reagents under mechanical action. The mechanochemical approach makes it possible to significantly accelerate the synthesis of LDHs and reduce the amount of alkaline wash water [1]. In some publications [25–28], the mechanochemical method for producing CuAl-LDHs is described. In refs. [26,27], a two-step method including mechanical activation and mixing in water was implemented using basic copper carbonate (malachite) and aluminum hydroxide (gibbsite). The variation of the Cu/Al molar ratio from 0.5 to 4.0 and the rotation speed of the balls in a mill from 200 to 600 rpm led to the formation of a pure Cu-Al LDH phase (at Cu/Al = 2 and 600 rpm). In ref. [28], a 10% aqueous suspension of malachite and aluminum hydroxide was treated in a ball mill. This grinding step was necessary for the faster formation of LDH, but for a more complete conversion of raw materials into the target product, an aging step was also necessary.

Our work is related to the development of methods for the CuAl-LDHs synthesis. It not only justifies the choice of conditions for obtaining the single-phase product by

co-precipitation and mechanochemical activation but also reveals the effect exerted by the synthesis method on the structure of CuAl-LDHs as well as on the formation and properties of catalytic compositions based on them. The effect of the catalyst's synthesis method on its activity and selectivity in the transformation of polyfunctional compounds is demonstrated using the liquid-phase hydrogenation of crotonaldehyde as an example. Selective hydrogenation of $\alpha,\beta$-unsaturated carbonyl compounds is an important reaction in the pharmaceutical, perfumery, and flavor industries [29]. As a rule, catalysts containing noble metals [29] and/or reduced at high temperatures [30] are used for this purpose. Therefore, the creation of catalysts containing a cheaper metal such as copper, whose reduction occurs at a low temperature, is an urgent task. The development of various reproducible methods for the synthesis of such catalysts will help to cope with this challenge.

## 2. Results

### 2.1. Synthesis of CuAl-LDH by the Co-Precipitation Method

Features of the $Cu^{2+}$ behavior in comparison with other $M^{2+}$ in the synthesis of LDH are caused by the nature of the copper cation. Since bivalent copper contains nine electrons in the 3d orbital, this is one of the reasons for its main crystallochemical property—the Jahn-Teller effect [31], which means that a distortion of the octahedral coordination structure leads to a gain in energy. Under certain conditions, for example, those favorable for the location of $Cu^{2+}$ cations in nearby octahedra, the formation of copper compounds with distorted octahedra (such as malachite ($Cu_2(CO_3)(OH)_2$) and rouaite ($Cu_2(NO_3)(OH)_3$)) becomes energetically preferable to the formation of LDH; therefore, when obtaining CuAl-LDH, detailed control of the synthesis parameters is required.

To choose the optimal pH for precipitation, an experiment was performed in which a precipitant solution (1M $Na_2CO_3$ and 3.5M NaOH) was gradually added to a solution containing aluminum and copper nitrates (50 mL 0.5M $Cu(NO_3)_2 \cdot 3H_2O$ and 50 mL 0.25M $Al(NO_3)_3 \cdot 9H_2O$) with simultaneous pH measurement. As a result, the dependence of the solution pH on the volume of the added precipitant was established (Figure 1a). In addition, at the fixed pH values (4.8, 8.0, 9.0, 10.5, and 13.0), samples of the precipitate formed under these conditions were taken to analyze the processes occurring in the different formation steps of the target CuAl-LDH. According to the XRD data (Figure 1b, line 1), at a lower pH value, the main phases were hydroxynitrate $Cu_2(NO_3)(OH)_3$ and hydroxycarbonate $Cu_2(CO_3)(OH)_2$. Starting at pH 8.0, LDH is formed in a significant amount, but the product is not a single-phase one, and a comparable amount of the rouaite phase is present. In the pH range of 9–10.5, a single-phase CuAl-LDH with the ratio Cu:Al = 2 was obtained, which had a set of reflections typical of the layered hydrotalcite-like phase (Figure 1b, line 2). With an increase in pH to 13, the structure of LDH is destroyed, and the formation of hydroxide and copper oxide is observed (Figure 1b, line 3).

Subsequent synthesis of CuAl-LDH in the carbonate form (containing $CO_3^{2-}$ anions in the interlayer space) was carried out using a typical co-precipitation procedure reported in [30,31]; solutions of aluminum and copper nitrates were added to an excess of $Na_2CO_3$ solution. However, maintaining the established optimal pH value of 9 did not ensure the formation of a single-phase product, and along with the target phase of LDH, the malachite phase was present in significant quantities (Figure 2a). As it was noted in some studies [13], in the synthesis of CuAl-LDH, not only the pH value is an important parameter but also the $CO_3^{2-}/Al^{3+}$ ratio. Thus, the transition from an excess of carbonate ions to $CO_3^{2-}/Al^{3+}$ ratios close to the stoichiometric 0.5 is necessary for the formation of the target product. Typical diffraction patterns of the single-phase CuAl-LDH obtained at pH 9 and $CO_3^{2-}/Al^{3+} = 0.76$ and 0.5 are displayed in Figure 2b.

Further experiments revealed that at the $CO_3^{2-}/Al^{3+}$ ratio of 0.5 and 0.76 and pH in the range of 9–10.5, the single-phase CuAl-LDH product is formed reproducibly regardless of the order of addition of reagents, the precipitation temperature, and the aging temperature of the mother liquor. At the same time, an increase in the aging temperature from 25 to 70 °C contributes to an increase in the crystallinity of the target CuAl-LDH, which is

reflected in an increase in the intensity of diffraction peaks with a decrease in their width (Figure S1, [11]).

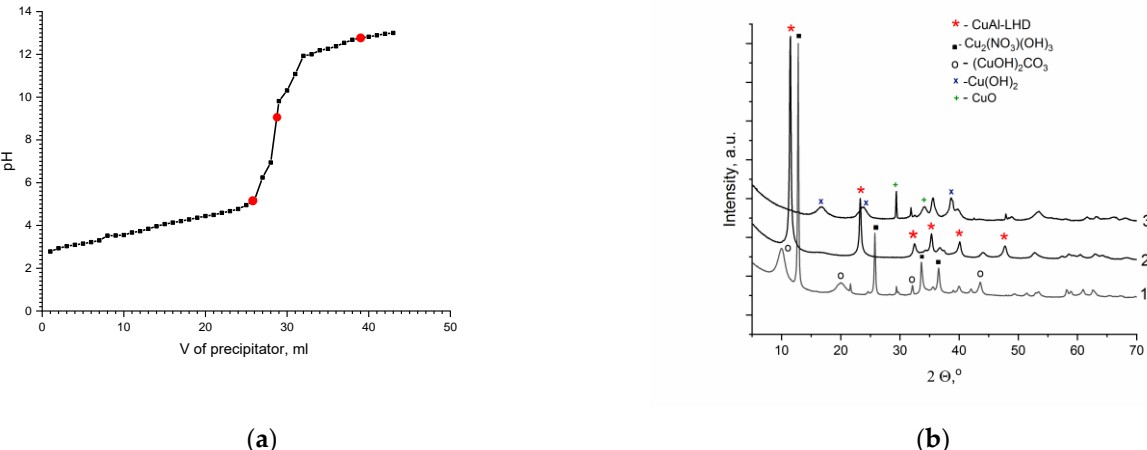

(**a**) (**b**)

**Figure 1.** Dependence of the pH value on the precipitant volume (**a**); diffraction patterns of the samples obtained at different pH values (**b**): 1—pH 4.8, 2—pH 9.0, and 3—pH 13.0.

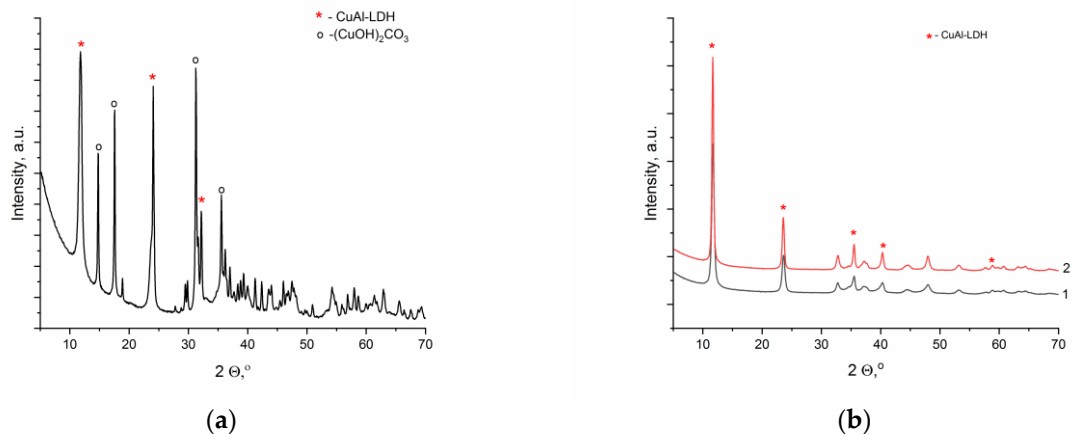

(**a**) (**b**)

**Figure 2.** Diffraction patterns of the samples synthesized at pH 9 and $CO_3^{2-}/Al^{3+}$ ratios equal to 4 (**a**), 0.76 ((**b**), line 1), and 0.5 ((**b**), line 2).

### 2.2. Synthesis of CuAl-LDH by Mechanochemical Method

In the mechanochemical synthesis of CuAl-LDH, a triple mixture of $Cu(OH)_2$, $Al(OH)_3$ (gibbsite), and $Na_2CO_3$ (which served as an available source of carbonate ions) was used, in contrast to the mixture of malachite and gibbsite described in the literature. To obtain LDH, a two-step process was chosen, which included dry grinding and subsequent aging of the mechanically activated product in water [1]. In mechanochemical synthesis, the variable parameters, along with the ratio of components, are the grinding time, the intensity of the impact (centripetal acceleration of milling bodies), as well as the aging time and temperature.

In the first step, the grinding time was varied in the range of 15–30 min using the maximum acceleration of milling bodies possible for the equipment used (the mass ratio of reagents to milling bodies of 1:20; the aging in a twenty-fold excess of distilled water at room temperature under constant stirring for 2 h). According to the literature data [32], high intensity of exposure is important for significant amorphization of the interacting components in the first step.

In the diffraction patterns shown in Figure 3, peaks at 2 θ equal to 11.73 and 23.62°, which are characteristic of the hydrotalcite-like phase, are observed already at the minimum

time of mechanical activation. However, with an increase in the duration of grinding, there is a significant decrease in the intensity of these peaks, with a simultaneous growth of the intensity of peaks in the range of 2 θ angles 35–45° corresponding to the copper oxide phase (CuO, PDF file No. 01-089-5895). Probably, due to local overheating of the reaction mixture during a prolonged high-energy exposure, dehydration of the hydroxide phase occurs with the formation of CuO. In this connection, further experiments aimed at optimizing the conditions of mechanochemical synthesis were carried out at a lower acceleration of milling bodies, 500 m·s$^{-2}$, varying the time of mechanical action in a wider range from 15 to 90 min (Figure 4).

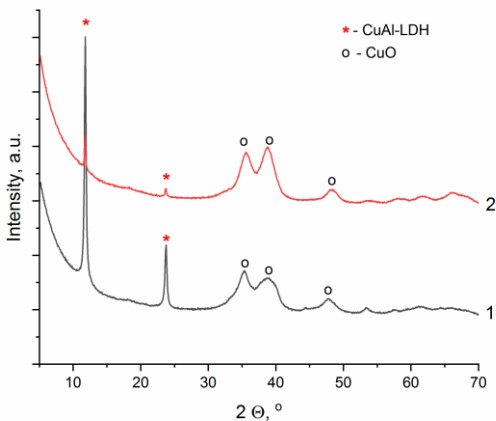

**Figure 3.** Diffraction patterns of the CuAl-LDH samples obtained by two-step mechanochemical synthesis during the mechanical activation step at 1000 m·s$^{-2}$: 1—for 15 min, 2—for 30 min (the mass ratio of reagents to milling bodies of 1:20; the aging in a twenty-fold excess of distilled water at room temperature under constant stirring for 2 h).

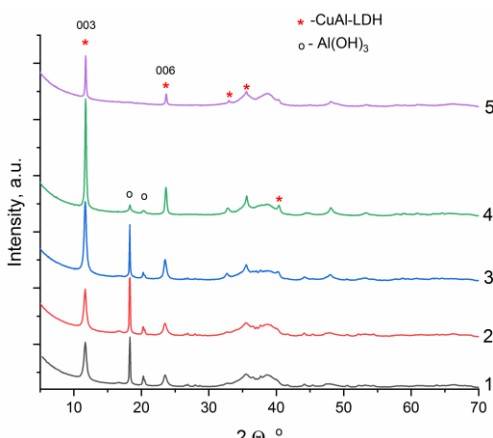

**Figure 4.** Diffraction patterns of the samples obtained by mechanochemical synthesis at 500 m·s$^{-2}$ using different activation times (1—15 min, 2—30 min, 3—45 min, 4—60 min, 5—90 min; the mass ratio of reagents to milling bodies of 1:20; the aging in a twenty-fold excess of distilled water at room temperature under constant stirring for 2 h).

With a minimum time of mechanical activation equal to 15 min, reflections of the CuAl-LDH phase are observed, but in the 2 θ range of 15–21°, there are intense peaks corresponding to unreacted aluminum hydroxide (Figure 4, line 1). As the time of mechanical activation is extended, the intensity of LDH peaks increases and that of aluminum hydroxide peaks decreases (Figure 4, lines 2–5). The time of mechanical action equal to 60 min can be considered optimal since trace amounts of Al(OH)$_3$ and the most intense peaks of the target product are observed. Although there is no impurity phase during grinding for



90 min and a single-phase LDH is observed, the intensity of its peaks (especially 003 and 006) decreases significantly, probably due to the amorphization of the LDH structure.

### 2.3. Investigation of the Characteristics of the Synthesized LDHs

To establish the effect of the synthesis method on the structure and properties of CuAl-LDHs, samples prepared under optimal conditions for the corresponding method with the same molar ratio of Cu/Al = 2 were selected. Co-precipitation (the sample denoted as CuAl-2-cp) was carried out at pH 9 and a $CO_3^{2-}/Al^{3+}$ ratio of 0.76; mechanochemical synthesis (the sample denoted as CuAl-2-ma) was performed at an acceleration of 500 m·s$^{-2}$ for 60 min. Both methods included the second step of precipitate aging in water.

Local EDX analysis of individual sections of the produced materials (Table S1) revealed a uniform distribution of elements with Cu/Al atomic ratios in the range of 1.9–2.3 and 2.2–2.3 for the samples obtained by co-precipitation and mechanochemical synthesis, respectively. Thus, the revealed values correspond to those specified and estimated by chemical analysis after complete dissolution of the samples.

The structural parameters of the CuAl-2-cp and CuAl-2-ma samples were compared using the diffraction patterns displayed in Figures 2b and 4. All the patterns have characteristic peaks at small diffraction angles corresponding to crystallographic planes (00l), which indicate the formation of a layered hydrotalcite-like structure. Symmetrical and intense reflections confirm the formation of a well-crystallized structure. It is known that the formed hydroxide layers can be packed in different ways, and the formation of various structural polytypes is possible [13]. To compare the obtained samples, at this stage of the work, the lattice constants *a* and *c* were refined using peaks (003), (006), and (110) within the conventional approach to analyzing the structure of hydrotalcite-like compounds (3R1 polytype) (PDF-2 file No. 00-037-0630).

It was found that, despite the fundamentally different approaches to the synthesis of samples, they have almost identical structural parameters (Table S2): parameter *c* is 2.261 and 2.268 nm, and parameter *a* is 0.304 and 0.305 nm for CuAl-2-cp and CuAl-2-ma, respectively.

According to the results of scanning electron microscopy, all the synthesized CuAl-LDH samples have a lamellar morphology with plates oriented in different directions (Figure 5). An analysis of the images obtained at the same magnification showed that the dimensions of the plates differ significantly and depend on the synthesis method. For samples obtained by the mechanochemical method, the size is much larger. Thus, the width of the plates varies between 44 and 85 nm, in contrast to CuAl-2 cp, where this range is 9–18 nm. The plate diameter is 500–850 nm and 220–350 nm for CuAl-2-ma and CuAl-cp, respectively.

### 2.4. The Formation of Catalysts Based on CuAl-LDH

The synthesis of copper-containing catalysts with CuAl-LDHs as a precursor includes high-temperature sequential steps of oxidative and reductive treatment. To elucidate the effect of the CuAl-LDH preparation method on the formation conditions of the oxide phase and active metal centers, thermal analysis and temperature-programmed reduction were used.

Differential weight loss curves in the temperature range of 30–700 °C for the studied samples are shown in Figure 6. Qualitatively, both curves are similar and demonstrate the three-step decomposition typical of LDHs [33–35], with a total weight loss of 29.5 and 24.6% for CuAl-2-cp and CuAl-2-ma, respectively. The low-temperature shoulder is associated with the presence of an insignificant amount of physically adsorbed water. In the temperature range up to 200 °C, interlayer water is removed. However, the temperature conditions of dehydroxylation of brucite-like layers and the removal of interlayer anions differ markedly depending on the synthesis method of CuAl-LDHs. In the case of the sample synthesized by the mechanochemical method, the formation of the oxide phase is almost complete at 300 °C. The sample obtained by co-precipitation contains a considerable

amount of strongly bound carbonates [11], the maximum removal rate of which is observed at 550 °C. Thus, the temperature required for the formation of the oxide phase differs depending on the method of LDH synthesis used. However, for a more correct comparison of the samples, the same calcination temperature of 550 °C was used for all precursors. The obtained phases of mixed oxides have a similar structure (Figure S2): the lattice parameter $a$ is equal to 0.512 and 0.511 nm, and parameter $c$ is 0.472 and 0.473 nm for $CuAlO_x$-2-cp and $CuAlO_x$-2-ma, respectively (Table S2).

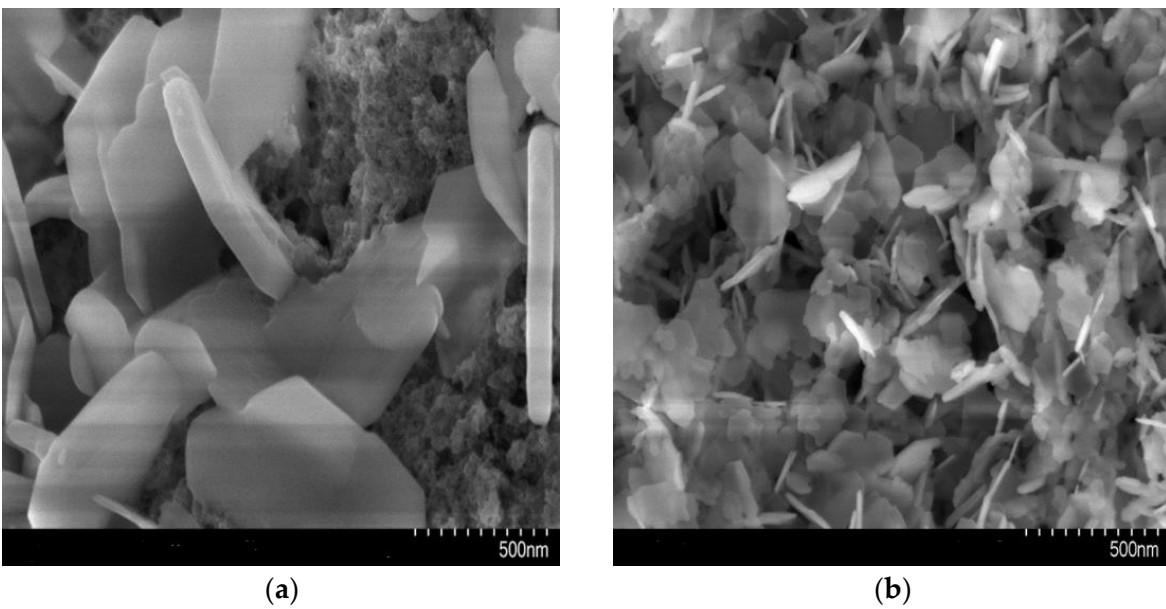

(**a**)                                                                                            (**b**)

**Figure 5.** SEM images of the samples obtained by (**a**) mechanochemical methods and (**b**) co-precipitation.

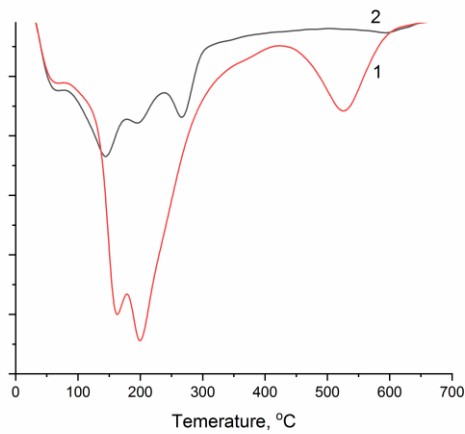

**Figure 6.** Thermograms of samples 1—CuAl-2-cp and 2—CuAl-2-ma obtained by co-precipitation and mechanical activation (before measurements, the samples were dried under vacuum at 25 °C).

The reduction conditions of metals were studied using the temperature-programmed reduction method (TPR) after preliminary calcination at 550 °C in air (Figure 7).

The TPR profiles of the studied samples have a similar shape, represented by two unresolved peaks. At the same time, for CuAl-2-ma, they are shifted to the low-temperature region, which may be caused by the smaller size of the crystallites formed during the mechanochemical synthesis ($L_c$ is equal to 23.5 and 45.9 nm for $CuAlO_x$-2-ma and $CuAlO_x$-2-cp, respectively). The presence of two hydrogen consumption regions with maxima at 193 and 246 °C for CuAl-2-cp and 189 and 200 °C for CuAl-2-ma may be associated with

the two-step reduction of copper: $Cu^{2+} \rightarrow Cu^+ \rightarrow CuO$ [35]. The presence of the additional high-temperature peak observed in the case of CuAl-2-cp is attributed to the reduction of larger CuO or $CuAl_2O_4$ particles [36]. The measurement of the hydrogen amount consumed during the TPR experiments made it possible to estimate the degree of copper reduction based on the stoichiometry of the $Cu^{2+} + H_2 = Cu^0 + 2H^+$ reaction; it was equal to 95% for the sample obtained by the mechanochemical method and 70% for the sample obtained by co-precipitation. Thus, although there are no hydrogen consumption signals on the TPR profile after 300 °C for both samples, a higher temperature is probably necessary for a deeper reduction of copper in the catalysts obtained by co-precipitation. It was also shown in ref. [35] that to obtain copper particles active in the hydrogenation of propyne, the catalyst should be reduced at a higher temperature. Thus, we used two reduction temperatures, 300 and 550 °C, when comparing the catalysts (the samples denoted as CuAl-ma-300(550) and CuAl-cp-300(550)).

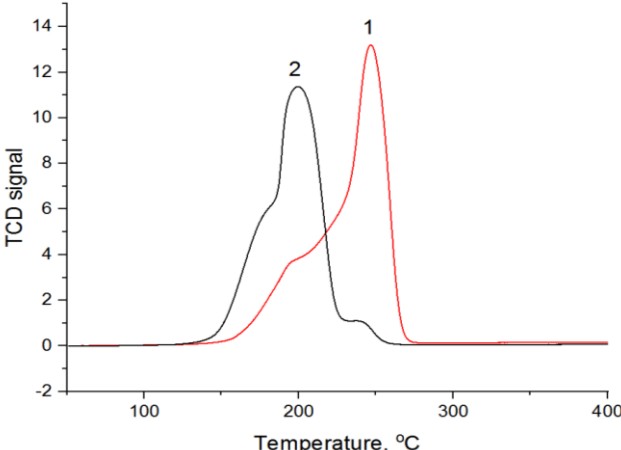

**Figure 7.** TPR profiles for 1—$CuAlO_x$-2-cp and 2—$CuAlO_x$-2-ma. Before measurements, CuAl-LDH samples were calcined at 550 °C.

The analysis of the XRD data showed that, unlike CuAl-ma-300, in the case of the sample obtained by co-precipitation, a temperature of 300 °C is really insufficient to obtain the same degree of copper reduction, and a significant proportion of copper in the oxidized state is present in the CuAl-cp-300 sample (Figure 8a), which is consistent with the degree of copper reduction calculated from the TPR experiment. An increase in the reduction temperature leads to an increase in the amount of metallic copper and/or its crystallinity in both pairs of samples (Figure 8a,b). The size of copper particles calculated from the diffraction peaks (111) at $2\theta = 43.32°$ increases almost twice during the transition from CuAl-ma-300 to CuAl-ma-550 and virtually does not change with an increase in the reduction temperature of the samples prepared from CuAl-cp (Table 1).

It should be noted that the presented diffraction patterns do not contain signals that could be attributed to aluminum-containing phases. This may be related to the much lower intensity of the signals compared to the intensity of reflections from elemental copper and copper oxide. In addition, this may be caused by the formation of an X-ray amorphous or poorly crystallized phase. At the same time, the electron microscopy study made it possible to reveal the presence of such a phase.

The TEM study revealed that all samples of the tested catalysts comprise conglomerates of particles with a size of 5–50 nm, which are enriched with aluminum and have a structure close to that of copper aluminate $CuAl_2O_4$ (Figure 9). Against the background of such conglomerates, in the TEM images of CuAl-ma-300(550) samples (Figure 9a–d), one can see numerous contrast particles of a globular shape with sizes in the range from 10 to 100 nm; according to EDS analysis, their chemical composition corresponds to copper. In comparison with other samples, CuAl-cp-300 has a much lower content of metallic copper

particles (Figure 9e,f). Copper oxide particles were detected in this sample. However, with increasing the reduction temperature, the formation of well-crystallized copper particles was observed (Figure 9g,h).

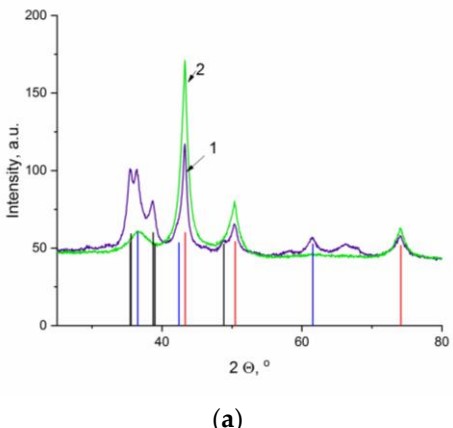

(**a**)

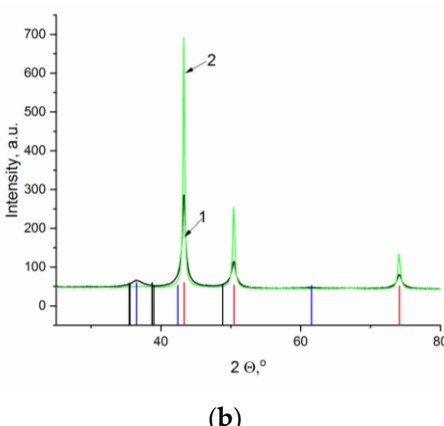

(**b**)

**Figure 8.** Diffraction patterns of the samples calcined at 550 °C and reduced at 300 °C (1) and 550 °C (2): (**a**) CuAl-cp and (**b**) CuAl-ma (red line—basic reflections of Cu, PDF file No. 01-085-1326; blue line—basic reflections of $Cu_2O$, PDF file No. 01-077-0199; black line—basic reflections of CuO, PDF file No. 01-080-0076).

**Table 1.** Textural characteristics of the studied samples and the CSR size of $Cu^0$.

| Sample | $S_{BET}$, $m^2$ $g^{-1}$ | $V_{ads}$, $cm^3$ $g^{-1}$ | CSR (111), Å |
|---|---|---|---|
| CuAl-ma-300 | 77 | 0.26 | 173 |
| CuAl-ma-550 | 53 | 0.21 | 308 |
| CuAl-cp-300 | 99 | 0.37 | 70 |
| CuAl-cp-550 | 71 | 0.30 | 81 |

Investigation of the textural characteristics of the obtained catalysts revealed the effect exerted by the synthesis method of the CuAl-LDH precursor on the specific surface area and structure of the pore space. Nitrogen adsorption-desorption isotherms for CuAl-ma-300(550) and CuAl-cp-300(550) samples belong to type II according to the IUPAC classification [37], and the presence of hysteresis loops in the region of $0.5 < P/P_0 < 1.0$ indicates the presence of mesopores and macropores. The shape of the loops corresponds to the H3 type and is typical of non-rigid aggregates of lamellar particles [37]. From the analysis of the pore size distribution curves (PSDC) calculated by the BJH method (insets in Figure 10a,b), it follows that the CuAl-ma-300(550) samples (an inset in Figure 10a) have a non-uniform pore size distribution in the region from 4 to 100 nm, with the maxima at the pore sizes 15 and 19 nm. The main contribution to the pore volume, up to 70%, is made by mesopores with sizes from 10 to 40 nm. The samples obtained by co-precipitation, CuAl-cp-300(550), have a wider pore size distribution, from 4 to 140 nm, and larger values of the maxima, 54 and 47 nm; their pore structure consists predominantly (up to 70%) of large meso- and macropores (an inset in Figure 10b). In the PSDC of these samples, a mesopore region is observed with a diameter of less than 8 nm, but the fraction of such mesopores does not exceed 6–8%.

With an increase in the reduction temperature from 300 to 550 °C, regardless of the precursor preparation method, there is a decrease in the proportion of pores with a smaller diameter and a corresponding decrease in the specific surface area (by 30%) and pore volume (by 20%) (Table 1).

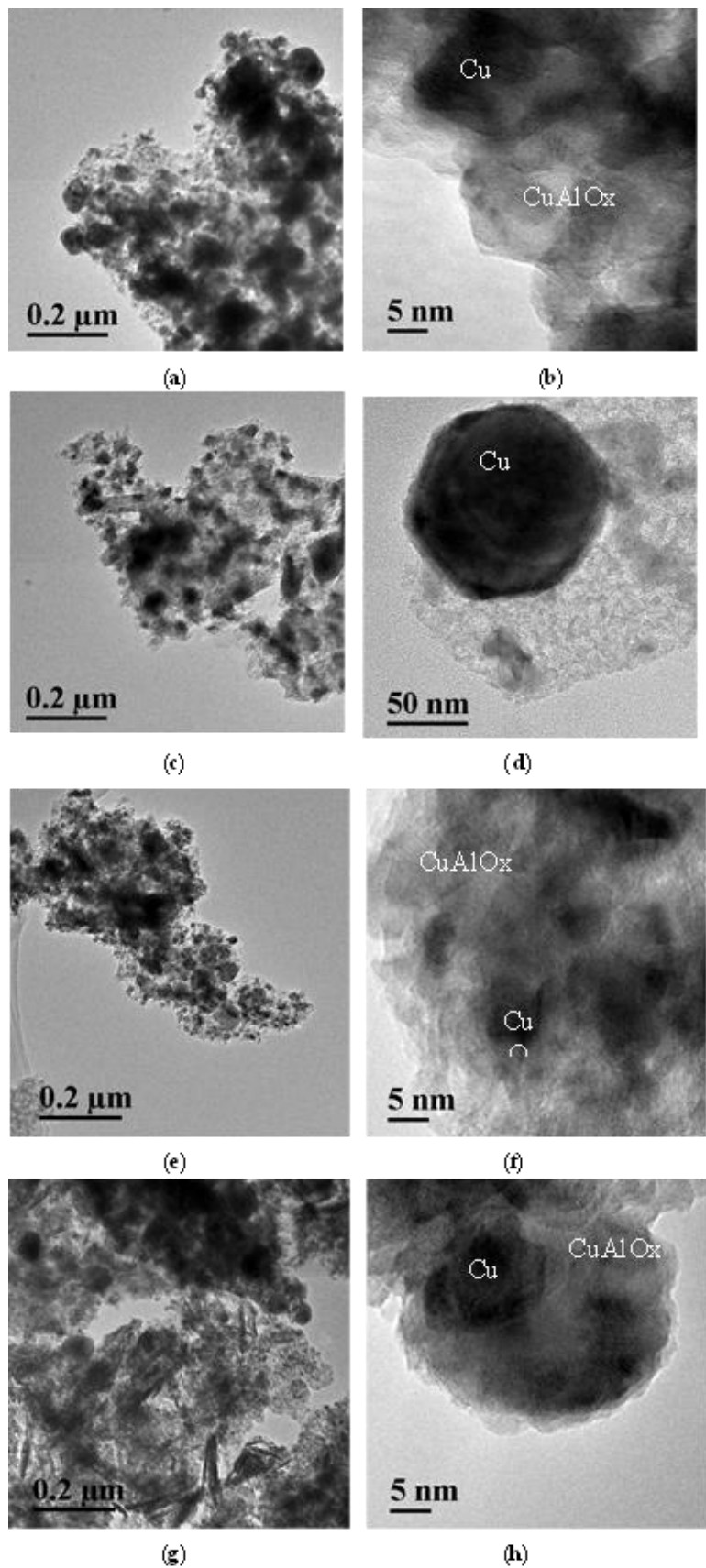

**Figure 9.** TEM images of samples (**a**,**b**) CuAl-ma-300, (**c**,**d**) CuAl-ma-550, (**e**,**f**) CuAl-cp-300, and (**g**,**h**) CuAl-cp-550.

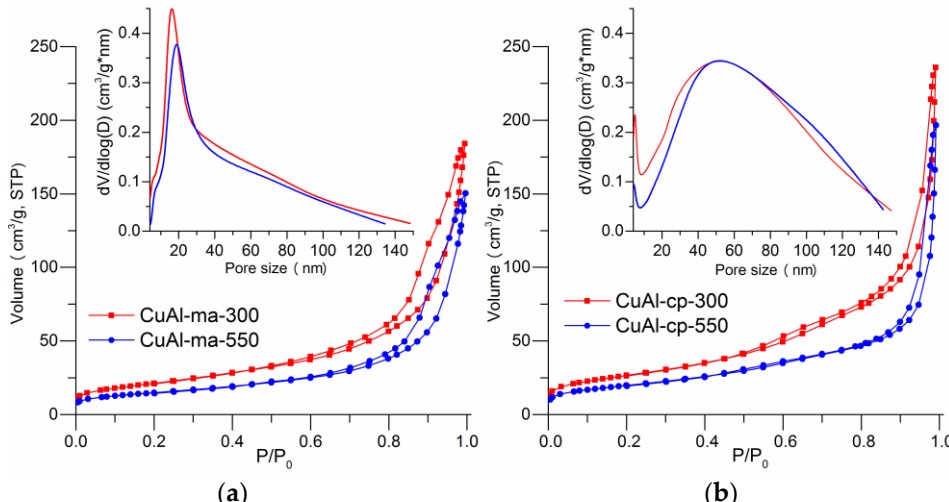

**Figure 10.** Nitrogen adsorption isotherms at 77 K for samples CuAl-ma-300 and CuAl-ma-550 (**a**); CuAl-cp-300 and CuAl-cp-550 (**b**). An inset: the BJH pore size distribution curves.

*2.5. Catalytic Properties in the Liquid-Phase Hydrogenation of Crotonaldehyde*

The obtained samples of CuAl-cp-300(550) and CuAl-ma-300(550) catalysts were studied in the reaction of crotonaldehyde (butenal) liquid-phase hydrogenation, which allowed us to establish the influence of the synthesis method and pretreatment conditions of the catalysts not only on their activity but also on their selectivity in the formation of individual products. According to the scheme of butanal transformation (Figure 11), along with the exhaustive hydrogenation yielding butanol, it is possible to obtain the products of selective hydrogenation of a double bond or a carbonyl group [38].

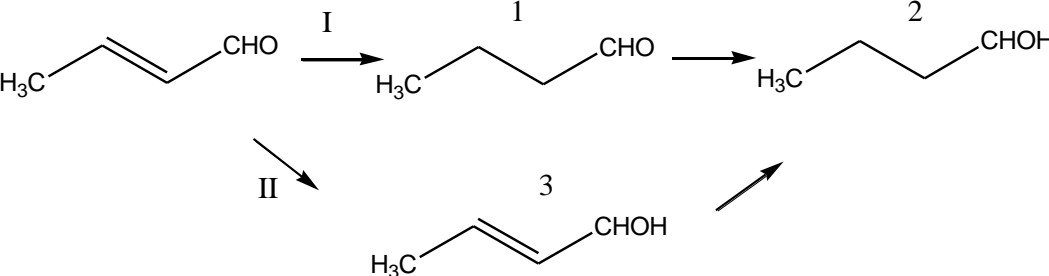

**Figure 11.** A scheme of crotonaldehyde hydrogenation: direction I—sequential hydrogenation to butanal (1) and then to butanol (2); direction II—hydrogenation of the aldehyde group to obtain 2-butene-1-ol (crotyl alcohol, 3) and then butanol (2).

A comparison of the catalysts reduced at 300 °C showed that the sample prepared from the LDH synthesized by mechanochemical method had greater activity: the conversion of butenal reached 99.9 and 50.5% for CuAl-ma-300 and CuAl-cp-300, respectively. The difference in activity may be determined by the lower proportion of reduced copper in CuAl-cp-300. At the same time, the properties of the active centers are similar, and the main direction of the reaction includes hydrogenation of the double bond with a butanal formation selectivity of 86 and 93% for CuAl-ma-300 and CuAl-cp-300, respectively. The higher activity of CuAl-ma-300 contributes to deeper hydrogenation with the formation of butanol (selectivity of 10%) (Figure 12a).

Despite the fact that an increase in the reduction temperature leads to an increase in the proportion of metallic copper, the activity of both catalysts decreases, and the conversion is 29 and 26% for CuAl-ma-550 and CuAl-cp-550, respectively. The decrease in activity is difficult to explain in terms of copper particles sintering with an increase in the reduction temperature since the particle size in CuAl-cp-300(550) virtually does not change. However,

in both samples, rearrangement of the pore space is observed, which is associated with a decrease in the fraction of small pores and the specific surface area. Probably, as a result of such rearrangement, a part of the active metal becomes inaccessible to reagents. It is essential that in this case, with a close conversion value, the selectivity of the catalysts be fundamentally different. The catalytic behavior of CuAl-cp-550 is similar to the samples reduced at 300 °C, and the main product is butanal. However, when CuAl-ma-550 is employed, the direction of the carbonyl group hydrogenation is implemented to a greater extent, and butenol is formed with a selectivity of 53%. Since the main distinction of the CuAl-ma-550 sample is the appearance of larger copper particles at a reduction temperature of 550 °C (Table 1), it can be assumed that this affects the adsorption interaction between butenal and the catalyst surface; the molecule is adsorbed on the Cu surface mainly as the η1-(O)-configuration, as a result of which the hydrogenation proceeds selectively by the C=O bond.

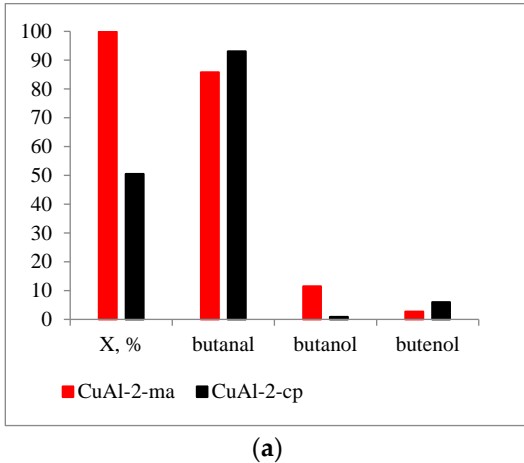

(**a**)

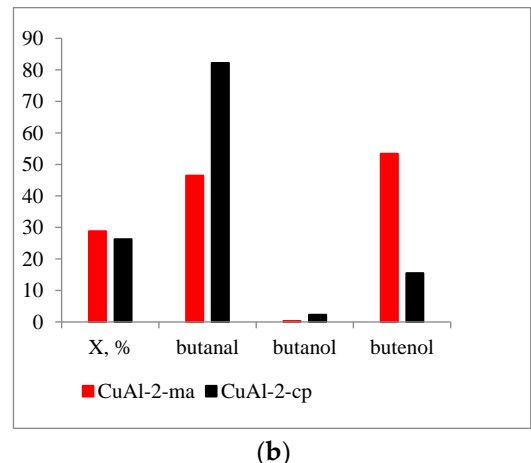

(**b**)

**Figure 12.** Dependence of conversion and selectivity of the crotonaldehyde hydrogenation on the chosen catalyst and reduction temperature: (**a**)—$T_{H2}$ = 300 ° and (**b**)—$T_{H2}$ = 550 ° (T = 80 °C, P = 0.5 MPa, 180 min, ethanol).

## 3. Materials and Methods

CuAl-LDHs were synthesized using aluminum nitrate $Al(NO_3)_3 \cdot 9H_2O$ (high-purity grade), copper nitrate $Cu(NO_3)_2 \cdot 3H_2O$ (analytical grade), aluminum hydroxide $Al(OH)_3$ (gibbsite), sodium carbonate $Na_2CO_3$ (reagent grade), nitric acid $HNO_3$, and sodium hydroxide NaOH (reagent grade) (the reagents were purchased from Omskreactiv).

In the synthesis of Cu-LDHs by co-precipitation, an aqueous solution containing $Cu(NO_3)_2$ and $Al(NO_3)_3$ (with the molar ratio $Cu^{2+}/Al^{3+}$ = 2) was added dropwise to the solution of sodium carbonate (1 mol·$L^{-1}$) under stirring at room temperature. During the synthesis, a constant pH value of 9 ÷ 10.5 was maintained by adding a 1 M NaOH solution. The synthesis was carried out for 2 h. After the addition of the entire solution of salts, the suspension was stirred for one hour at synthesis temperature. To increase the crystallinity of the precipitate, its aging was carried out at room temperature and 70 °C for 18 h. The resulting product was washed with the neutral reaction of wash water, separated on a Schott filter, and left to dry in the air.

The mechanochemical synthesis of CuAl-LDHs was carried out in two steps. In the first step, mechanical activation was performed in a high-energy centrifugal ball mill of the planetary type AGO-2 (Novosibirsk Test Center Ltd., Novosibirsk, Russia). $Cu(OH)_2$, $Al(OH)_3$ (gibbsite), and $Na_2CO_3$ were used as the initial reagents. To obtain the molar ratio $Cu^{2+}/Al^{3+}$ = 2, a mixture of dry reagents was placed in steel drums with steel balls (8 mm in diameter) that served as the milling bodies. After that, mechanical activation was conducted in air with a mass ratio of reagents to milling bodies of 1:20; the activation time and centripetal acceleration of milling bodies varied between 15 and 90 min and

$500–1000 \ \mathrm{m \cdot s^{-2}}$. In the second step, the samples after mechanical activation were aged in a twenty-fold excess of distilled water at room temperature under constant stirring for 2 h. The samples obtained were filtered, washed with distilled water, and dried in the air for 12 h.

X-ray diffraction analysis of the samples was carried out on a Bruker D8 Advance diffractometer (Bruker, Billerica, MA, USA) with a CuKα source. Data were recorded in the 2 θ range of 5–80° with a step size of 0.05° and a counting time of 1 s per step. Signals were detected using a multichannel LynxEye detector. Phase analysis was performed by comparing the interlayer distances and intensities of corresponding reflections with theoretical values from the ICDD PDF-2 database.

The local composition of LDH was investigated by scanning electron microscopy (SEM) on a JSM-6460 LV (JEOL, Tokyo, Japan) instrument with a tungsten cathode and an accelerating voltage of 20 kV. TEM images were recorded on a JEM-2100 (JEOL, Tokyo, Japan) electron microscope with a lattice resolution of 0.14 nm at an accelerating voltage of 200 kV. The EDX analysis was performed using an INCA 250 spectrometer (Oxford Instruments High, Wycombe, UK). To obtain high-resolution images of the surface, a Regulus SU8230 FE-SEM scanning electron microscope (Hitachi, Tokyo, Japan) was used.

The metal content in the samples was measured by atomic absorption spectroscopy on an AA-6300 Shimadzu (Kyoto, Japan) spectrometer after dissolving 10 mL of $H_2SO_4$ (1:2) under heating. Quantitative determination of the Cu and Al content was performed under the following measurement conditions: Cu-atomic absorption mode, air-acetylene flame ($14.8 \ \mathrm{L \cdot min^{-1}}$–$2.0 \ \mathrm{L \cdot min^{-1}}$), wavelength 324.8 nm; Al-atomic absorption mode, nitrous oxide-acetylene flame ($10.0 \ \mathrm{L \cdot min^{-1}}$–$8.7 \ \mathrm{L \cdot min^{-1}}$), and wavelength 309.3 nm.

The thermal decomposition of the LDH samples was studied by means of thermal analysis (TG-DTG-DTA). The measurements were made on an STA-449C Jupiter (Netzsch-Gerätebau GmbH, Selb, Germany) instrument in dynamic mode in an air medium at a heating rate of $10 \ \mathrm{°C \cdot min^{-1}}$.

Temperature-programmed reduction (TPR) of the tested samples (after calcination at 550 °C for 2 h) was performed on an AutoChem II 2920 (Micromeritics, Norcross, GA, USA) chemisorption analyzer equipped with a highly sensitive thermal conductivity detector. The reduction was carried out using a calibrated mixture of 10 vol.% $H_2$ in argon; the temperature range was 35–500 °C; and a measuring cell with the sample was heated at a rate of $10 \ \mathrm{°C \cdot min^{-1}}$. The flow rate of the gas mixture through the reactor with the sample was $30 \ \mathrm{cm^3}$ (STP)$\cdot \mathrm{min^{-1}}$.

The specific surface area and pore structure characteristics of the samples were determined from the analysis of nitrogen adsorption-desorption isotherms at 77.2 K, which were measured on an ASAP-2020 volumetric vacuum static setup (Micromeritics, Norcross, GA, USA). Before the adsorption measurements, all the samples were evacuated at 300 °C for 6 h. The equilibrium relative pressures of the nitrogen adsorption-desorption isotherms ranged from $10^{-3}$ to $0.996 \ P/P_0$. The BET specific surface area ($S_{BET}$) was calculated using the adsorption isotherm in the range of equilibrium relative pressures of nitrogen vapors ($P/P_0 = 0.05–0.25$). In the calculation, the molecular site of nitrogen in the filled monolayer was assumed to be $0.162 \ \mathrm{nm^2}$. The adsorption pore volume ($V_{ads}$) was determined from the nitrogen adsorption at $P/P_0 = 0.990$, assuming that the adsorbate density is equal to the density of the normal liquid, $0.808 \ \mathrm{g/cm^3}$. Pore size distribution curves (PSDC) were obtained from desorption branches of nitrogen adsorption isotherms by the BJH method.

The hydrogenation experiments were performed using a 5% solution of crotonaldehyde in ethanol. The reaction was conducted at a pressure of 0.5 MPa and a temperature of 80 °C for 180 min on a laboratory setup consisting of an autoclave with a magnetic stirrer and an EL-FLOW Select hydrogen flow regulator. The weight of the catalyst sample was 1.0 g, and the stirring rate was 1300 rpm.

The quantitative composition of the reaction mixture was determined on a Chromos GC-1000 (Chromos, Russia) gas chromatograph. The obtained chromatograms were processed using the Chromos software. Chromatographic analysis was performed on

a capillary column DB-1 (dimethylsiloxane as the stationary liquid phase; L = 100 m; d = 0.25 mm) with a flame ionization detector. Concentrations were calculated using the internal normalization method.

The conversion of 2-butenal $X$ (%) was calculated by the equation:

$$X = (c_{react}/c_{init}) * 100\%,$$

where $c_{react}$ is the concentration of the reacted 2-butenal and $c_{init}$ is the initial concentration of 2-butenal.

The selectivity for each product $S$ (%) was calculated as:

$$S_x = (c_x/\Sigma c) * 100\%$$

where $c_x$ is the concentration of $x$ substance and $\Sigma c$ is the total concentration of the reaction products.

## 4. Conclusions

The study performed showed the possibility of obtaining phase-homogeneous aluminum-copper LDHs by fundamentally different methods, provided that the necessary synthesis parameters are met. When using the co-precipitation method, the pH values (9–10.5) and the $CO_3^{2-}/Al^{3+}$ molar ratio (0.5–0.8) are the key parameters and allow avoiding the formation of copper hydroxynitrate and hydroxycarbonate.

A triple mixture of the reagents $Cu(OH)_2$, $Al(OH)_3$ (gibbsite), and $Na_2CO_3$ (as an available source of carbonate ions) was used for the mechanochemical synthesis of CuAl-LDHs. In the mechanochemical step, the intense energy exposure destroys the LDH structure, and the necessary decrease in acceleration of milling bodies should be compensated by an increase in the exposure time (up to 60 min). However, even with such an increase in the time of the first step, the mechanochemical method can significantly decrease the duration of the CuAl-LDH synthesis and the amount of alkaline wash water. In addition, CuAl-ma showed greater activity in oxidative and reductive reactions, thus allowing the formation of the oxide phase and metallic copper particles under milder conditions. The resulting catalyst demonstrated high hydrogenating activity in the conversion of unsaturated aldehyde (butenal) and provided selective hydrogenation of the C=C bond. It was found that an increase in the size of copper particles (in this work, it was achieved by increasing the reduction temperature) makes it possible to regulate the reaction selectivity and implement to a greater extent the direction of selective hydrogenation of the C=O bond.

**Supplementary Materials:** The following supporting information can be downloaded at: https://www.mdpi.com/article/10.3390/inorganics11060247/s1. Figure S1: Diffraction patterns of the sample synthesized by co-precipitation at the $CO_3^{2-}/Al^{3+}$ ratio equal to 0.76, pH 9, at different aging temperatures; Table S1: Results of local EDX analysis (by scanning electron microscopy); Table S2: Microstructural parameters of CuAl-LDHs and CuAlO$_x$ mixed oxides; Figure S2: Diffraction patterns of the samples calcined at 550 °C (1) obtained by co-precipitation and (2) mechanical activation (blue line: basic reflections of CuO, PDF file No. 01-080-0076).

**Author Contributions:** Conceptualization, O.B.B., E.N.T. and V.A.L.; methodology, E.N.T.; validation, O.B.B. and V.A.L.; formal analysis, O.V.G., I.V.M. and M.V.T.; investigation, E.N.T.; data curation, E.N.T., O.V.G., A.N.S. and M.V.T.; writing—original draft preparation, E.N.T.; writing—review and editing, O.B.B.; visualization, O.B.B. and V.A.L.; supervision, O.B.B. and V.A.L.; project administration, O.B.B. and V.A.L. All authors have read and agreed to the published version of the manuscript.

**Funding:** This work was supported by the Ministry of Science and Higher Education of the Russian Federation within the governmental order for the Boreskov Institute of Catalysis (project AAAA-A21-121011490008-3). The studies were carried out using the facilities of the shared research center "National Center of Investigation of Catalysts" at Boreskov Institute of Catalysis and the Omsk Regional Center of Collective Usage, Siberian Branch of the Russian Academy of Sciences.

**Data Availability Statement:** Data are contained in the article and Supplementary Materials. Any additional data is available on request from the corresponding author.

**Acknowledgments:** The authors thank Kirill E. Grudin for his help in conducting experiments, Tatiana I. Gulyaeva, Anastasiya V. Vasilevich, and Rinat R. Izmailov for their participation in the study of the samples.

**Conflicts of Interest:** The authors declare no conflict of interest.

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
