# Peer review of "Synthesis of CuAl-LDHs by Co-Precipitation and Mechanochemical Methods and Selective Hydrogenation Catalysts Based on Them"

_inorganics, doi:10.3390/inorganics11060247_

Round 1

Reviewer 1 Report

The authors prepared CuAl-LDHs catalysts through Co-Precipitation and Mechanochemical Methods. The differences of synthesis process and performance between the two methods are described in detail. The 2-butenal conversion (T = 80 oC, P = 0.5 MPa, 180 min, ethanol) for the sample from mechanochemical methods was 99.9%, in contrast to 50.5% for the catalyst obtained by co-precipitation. I would suggest its acceptance for publication in Inorganics after a major revision.

1. The Figures need to be further strengthened. e.g. the XRD patterns should added vertical coordinates, (intensity (a.u.)). There is an extra b in Figure 10.

2.  In the process of catalyst preparation, the author has relatively complete parameter optimization, but in the process of catalyst application research, it is insufficient. The authors should optimize some response parameters to make the results more reliable.

3. The author's analysis of the reasons for the differences in the performance of the two preparation methods is too superficial and needs to further improve the in-depth understanding.

4. Some of the latest preparation and catalytic applications of LDHs catalysts should be  Introduction in the preface. Please add more related papers (Nanomaterials 2022, 12(7), 1227; Fuel, (2023) 332,  126227);

5. Please clearly state the importance of the topic, recent examples from the literature. In the last paragraph of the Introduction, please state your motivation and the novelty of this research, and contribution to the literature.

Moderate editing of English language

Author Response

Reviewer 1:

Our general response: The authors thank the Reviewer for a careful analysis of the presented material. We checked and edited the text of the manuscript and tried to take into account all the recommendations

The authors prepared CuAl-LDHs catalysts through Co-Precipitation and Mechanochemical Methods. The differences of synthesis process and performance between the two methods are described in detail. The 2-butenal conversion (T = 80 ℃, P = 0.5 MPa, 180 min, ethanol) for the sample from mechanochemical methods was 99.9%, in contrast to 50.5% for the catalyst obtained by co-precipitation. I would suggest its acceptance for publication in Inorganics after a major revision.

1. The Figures need to be further strengthened. e.g. the XRD patterns should added vertical coordinates, (intensity (a.u.)). There is an extra b in Figure 10.

Thank you for your comments, the changes have been made to figures 1b, 2, 3, 4, 8.

2. In the process of catalyst preparation, the author has relatively complete parameter optimization, but in the process of catalyst application research, it is insufficient. The authors should optimize some response parameters to make the results more reliable.

Thank you for this comment. Indeed, the goal of this work was to optimize the synthesis of LDH by various methods, and this goal was achieved. As the reviewer correctly pointed out, the catalytic part is very concise. The purpose of the catalytic test at this stage of the work was only to demonstrate the presence of the hydrogenating activity of copper in the composition of these catalysts. Therefore, we chose relatively mild identical conditions to compare the catalytic properties of the obtained samples. In addition, we have already carried out a variation of the reaction conditions using these catalysts for the hydrogenation of a number of carbonyl compounds. This is a rather voluminous work, which will be presented in a separate article.

3. The author's analysis of the reasons for the differences in the performance of the two preparation methods is too superficial and needs to further improve the in-depth understanding.

There are many studies, including reviews, which describe in detail the physicochemical foundations and details of the synthesis of LDH by coprecipitation and mechanochemical methods. However, there are some peculiarities and problems in the synthesis of copper-containing LDHs by these methods. These are the questions we tried to address in this paper. Of course, in the future, I believe, our knowledge and understanding will be deeper, however, the current task has been solved. We have optimized the synthesis methods and compared some characteristics of samples of the same composition obtained by fundamentally different approaches.

4. Some of the latest preparation and catalytic applications of LDHs catalysts should be Introduction in the preface. Please add more related papers (Nanomaterials 2022, 12(7), 1227; Fuel, (2023) 332, 126227);

Thank you for your recommendations. We have expanded the introduction and added references 9 and 10.

5. Please clearly state the importance of the topic, recent examples from the literature. In the last paragraph of the Introduction, please state your motivation and the novelty of this research, and contribution to the literature.

Thank you for your advice. We have slightly reformulated the introduction to emphasize our motivation and the novelty of the study. According to your remarks English has been further edited by a highly qualified specialist. Corrections have been highlighted.

Reviewer 2 Report

This article entitled “Synthesis of CuAl-LDHs by Co-Precipitation and Mechanochemical Methods and Selective Hydrogenation Catalysts Based on Them” by Olga B. Belskaya et al. is suitable for Inorganics. The authors adequately compare the use of based CuAl-LDHs, prepared by co-precipitation and by mechanochemical methods, as catalysts for liquid-phase hydrogenation of crotonaldehyde, although the role of unreduced copper species in the catalysts, should also be considered in the discussion of the catalytic results.

1. Introduction section: The use of crotonaldehyde as a model substrate to study the hydrogenation of unsaturated aldehydes to obtain unsaturated alcohols should be mentioned in the introduction section.

2. 2.1. Synthesis of CuAl-LDH by co-precipitation method. The authors optimize different reaction parameters to obtain a single-phase CuAl-LDH by co-precipitation method. In the optimization of the CO32-/Al3+ ratio, the diffraction pattern of the sample synthesised at the CO32-/Al3+ ratio equal to 0.5, could be included in Figure 2(b) for a better comparison.

3. 2.2. Synthesis of CuAl-LDH by mechanochemical method. The authors indicate that the variable parameters are the ratio of components, the grinding time, the intensity of the impact and the aging time and temperatures. But they only studied the grinding time and the intensity of the impact. They should indicate throughout the text and in the figure captions, the values of the parameters that have been kept constant.

4. In addition, Figure 4, where five diffraction patterns have been represented, should be modified so that the peaks do not overlap and the evolution of the different phases over time can be clearly observed.

5. The section number 2.3. was duplicated.

6. The authors found that the temperature of dihydroxylation of LDHs differed with the synthesis method (300ºC for mechanochemical method and 550ºC for co-precipitation one), but both materials were calcined at the same temperature (550ºC). This could be a possible cause of the different reducibility of these materials. The authors should take this fact into account in their discussion.

7. The different reducibility of these materials was associated with the smaller particle size of the oxides obtained by mechanochemistry. The authors have indicated the size of the crystallites but, the SEM images of both oxides could also be included as supplementary material.

8. It might also be interesting for this discussion, to include the diffraction patterns of the calcined samples. These could be included in Figure 8 in order to compare the evolution of the diffraction signals after reduction at 300 and 550ºC.

9. Figure 9 should clearly identify which TEM images correspond to samples reduced at 300 or 550ºC.

10. In Figure 12 the reaction conditions should be included.

11. It has been described in the literature that Lewis acid sites facilitate the anchoring of crotonaldehyde molecule through its carbonyl double bond, facilitating the formation of 2-butene-1-ol. Thus, Lewis acidity is closely associated with the selectivity of the hydrogenation products of the C=O bond. Authors should consider the acid-base properties of the catalysts and discuss the synergistic effect of the presence of Lewis acid centers for carbonyl adsorption and metal sites for hydrogenation of the carbonyl bond of aldehyde. Thus, the presence of unreduced Cu2+ species could be considerate to explain the changes in conversion and selectivity of these catalysts reduced at 300 or 550ºC.

Comments and suggestions have also been attached.

Author Response

Reviewer 2:

Our general response: We are grateful to the reviewer for his/her positive overall evaluation of our article. We tried to take into account all the comments and, in accordance with them, made changes into the manuscript.

This article entitled “Synthesis of CuAl-LDHs by Co-Precipitation and Mechanochemical Methods and Selective Hydrogenation Catalysts Based on Them” by Olga B. Belskaya et al. is suitable for Inorganics. The authors adequately compare the use of based CuAl-LDHs, prepared by co-precipitation and by mechanochemical methods, as catalysts for liquid-phase hydrogenation of crotonaldehyde, although the role of unreduced copper species in the catalysts, should also be considered in the discussion of the catalytic results.

1. Introduction section: The use of crotonaldehyde as a model substrate to study the hydrogenation of unsaturated aldehydes to obtain unsaturated alcohols should be mentioned in the introduction section.

The authors thank the referee for the advice. We have added the required information in the Introduction, which is confirmed by the relevant references 29, 30.

2. 2.1. Synthesis of CuAl-LDH by co-precipitation method. The authors optimize different reaction parameters to obtain a single-phase CuAl-LDH by co-precipitation method. In the optimization of the CO32-/Al3+ ratio, the diffraction pattern of the sample synthesised at the CO32-/Al3+ ratio equal to 0.5, could be included in Figure 2(b) for a better comparison.

Thank you for the suggestion, additions have been made to fig. 2b.

3. 2.2. Synthesis of CuAl-LDH by mechanochemical method. The authors indicate that the variable parameters are the ratio of components, the grinding time, the intensity of the impact and the aging time and temperatures. But they only studied the grinding time and the intensity of the impact. They should indicate throughout the text and in the figure captions, the values of the parameters that have been kept constant.

Thank you for the helpful note; additions made and highlighted in yellow.

4. In addition, Figure 4, where five diffraction patterns have been represented, should be modified so that the peaks do not overlap and the evolution of the different phases over time can be clearly observed.

The Figure 4 has been corrected to make it clearer.

5. The section number 2.3. was duplicated.

This is our mistake, the numbering has been corrected.

6. The authors found that the temperature of dihydroxylation of LDHs differed with the synthesis method (300ºC for mechanochemical method and 550ºC for co-precipitation one), but both materials were calcined at the same temperature (550ºC). This could be a possible cause of the different reducibility of these materials. The authors should take this fact into account in their discussion.

We fully agree with the reviewer and understand that the calcination temperature 550ºC might not be optimal for the sample obtained by the mechanochemical method. In the future, when we synthesize a number of samples by this method, we will study the effect of this factor on the size of copper particles, their activity and selectivity in the hydrogenation of polyfunctional compounds. However, in the framework of this work, where it was important to compare the catalysts obtained by different methods, we considered that it would be more correct to fix all the conditions of thermal activation. We noted this in the text (page 7).

7. The different reducibility of these materials was associated with the smaller particle size of the oxides obtained by mechanochemistry. The authors have indicated the size of the crystallites but, the SEM images of both oxides could also be included as supplementary material.

Thanks for your valuable recommendation. We will certainly take this into account in the future, but we do not currently have SEM images of calcined samples. However, according to XRD data, even at a higher (possibly non-optimal) calcination temperature of 550°C, oxide phase СuAlOx-2-ma particles have a smaller size. We have added this information (CSR, nm) to Table S2.

8. It might also be interesting for this discussion, to include the diffraction patterns of the calcined samples. These could be included in Figure 8 in order to compare the evolution of the diffraction signals after reduction at 300 and 550ºC.

Thanks to the reviewer for the recommendation. Additional diffraction patterns of oxide phases are given in Supplementary Materials, Fig. S2.

9. Figure 9 should clearly identify which TEM images correspond to samples reduced at 300 or 550ºC.

Thank you for your helpful note; additions are made in the caption to Fig. 9

10. In Figure 12 the reaction conditions should be included.

Thank you for your helpful note; additions are made in the caption to Fig. 12

11. It has been described in the literature that Lewis acid sites facilitate the anchoring of crotonaldehyde molecule through its carbonyl double bond, facilitating the formation of 2-butene-1-ol. Thus, Lewis acidity is closely associated with the selectivity of the hydrogenation products of the C=O bond. Authors should consider the acid-base properties of the catalysts and discuss the synergistic effect of the presence of Lewis acid centers for carbonyl adsorption and metal sites for hydrogenation of the carbonyl bond of aldehyde. Thus, the presence of unreduced Cu2+ species could be considerate to explain the changes in conversion and selectivity of these catalysts reduced at 300 or 550ºC.

The reviewer outlined the fundamental and important issues. And we are well aware of the need to analyze and take into account these characteristics. However, we are not sure that their analysis is necessary in this article, which is more devoted to the features of the synthesis of copper-containing LDH and the conditions of catalyst formation. Since the samples obtained by different methods were compared in this work, their identical elemental composition suggests that their acid-base properties will also be similar. The presence of a part of copper in the oxidized state led to a decrease in conversion, but the selectivity of the reaction did not change; therefore, at this stage of the work, it can be assumed that the degree of participation of oxide forms of copper in the reaction is low. Undoubtedly, we will take into account all these issues in our further catalytic studies of these materials.
